# Regulation of Microglia-Activation-Mediated Neuroinflammation to Ameliorate Ischemia-Reperfusion Injury via the STAT5-NF-κB Pathway in Ischemic Stroke

**DOI:** 10.3390/brainsci12091153

**Published:** 2022-08-29

**Authors:** Zhijun Pu, Shengnan Xia, Pengfei Shao, Xinyu Bao, Dan Wu, Yun Xu

**Affiliations:** 1Department of Neurology, Nanjing Drum Tower Hospital, The Affiliated Hospital of Medical School of Nanjing University, Nanjing 210008, China; 2Institute of Brain Sciences, Nanjing University, Nanjing 210093, China; 3Jiangsu Key Laboratory for Molecular Medicine, Medical School of Nanjing University, Nanjing 210008, China; 4Jiangsu Province Stroke Center for Diagnosis and Therapy, Nanjing 210008, China; 5Nanjing Neurology Clinic Medical Center, Nanjing 210008, China

**Keywords:** dauricine, STAT5, NF-κB, inflammation, ischemia-reperfusion injury

## Abstract

Inflammatory reaction after ischemia-reperfusion contributes significantly to a worsened prognosis, and microglia activation is the main resource of inflammation in the nervous system. Targeting STAT5 has been shown to be a highly effective anti-inflammatory therapy; however, the mechanism by which the STAT5 signaling pathway regulates neuroinflammation following brain injury induced by ischemia-reperfusion remains unclear. Dauricine is an effective agent in anti-inflammation and neuroprotection, but the mechanism by which dauricine acts in ischemia-reperfusion remained unknown. This study is the first to find that the anti-inflammation mechanism of dauricine mainly occurs through the STAT5-NF-κB pathway and that it might act as a STAT5 inhibitor. Dauricine suppresses the inflammation caused by cytokines Eotaxin, KC, TNF-α, IL-1α, IL-1β, IL-6, IL-12β, and IL-17α, as well as inhibiting microglia activation. The STAT5b mutant at Tyr-699 reverses the protective effect of dauricine on the oxygen-glucose deprivation-reperfusion injury of neurons and reactivates the P-NF-κB expression in microglia. These results suggest that dauricine might be able to suppress the neuroinflammation and protect the neurons from the injury of post-ischemia-reperfusion injury via mediating the microglia activation through the STAT5-NF-κB pathway. As a potential therapeutic target for neuroinflammation, STAT5 needs to be given further attention regarding its role in ischemic stroke.

## 1. Introduction

Inflammation is closely related to both injury and repair processes after stroke, and has been suggested to play a causal role in the pathogenesis of stroke [1]. Persistent inflammation is associated with poor prognosis, and a higher baseline level of inflammation predicts short-term poor outcomes in large stroke [2]. Microglia are the main resident immunoreactive cells in the brain, play an important role as the first line of defense against central nervous system injury and inflammation, and can be activated within minutes of ischemic brain injury [3,4]. The identification of early recurrent stroke was improved using the symptomatic carotid atheroma inflammation lumen-stenosis score [5], and plaque inflammation-related f-fluorodeoxyglucose uptake was showed to independently predict future recurrent stroke post-PET [6]. The clinical outcomes after acute ischemic stroke could be improved by targeting immunity and inflammation [7].

Dauricine (C_38_H_44_N_2_O_6_) is a monomer derived from Chinese traditional herbs with rich pharmacological activity, especially in anti-inflammation and neuroprotection. By inhibiting the NF-κB pathway, dauricine may be able to reduce endothelial inflammation [8], inhibit the inflammatory response generated by LPS or cecal ligation and puncture [9], and inhibit severe pneumonia co-infection with streptococcus pneumoniae combined with clindamycin [10]. Dauricine plays a neuroprotective role mainly through upregulating GPX4 expression to inhibit the ferroptosis of nerve cells after intracerebral hemorrhage [11], inhibiting the inflammatory process after focal cerebral ischemia/reperfusion [12], and inhibiting HERG-encoded potassium channels [13]. However, while it is known that dauricine plays a role in anti-inflammation, the anti-inflammation target of dauricine remains unknown.

In this study, we found a new anti-inflammation mechanism of dauricine-theSTAT5-NF-κB pathway. Acting as a STAT5 inhibitor, dauricine not only reduced the inflammatory reaction induced by LPS or ischemia-reperfusion injury through regulating microglia activation, but also rescued neurons from injury caused by post-ischemia-reperfusion in an ischemia stroke mouse model.

## 2. Materials and Methods

### 2.1. Cell Culture

As described previously, 24-h-old C57BL/6J mice were used to isolate and purify primary microglial cells [14]. The cortical neurons were isolated from C57BL/6J mice embryos at the E15–17 stage [15]. We will not describe this in detail. The microglial cells were acquired and placed onto the indicated plates with new complete medium, and primary neurons were cultured in medium containing B27 (Life Technologies, B27 supplement 50 × 17,504, Grand Island, NY, USA) and glutamine (Gibco, GlutaMAX^TM^ 200 Mm (100×), 35050061, Grand Island, NY, USA). The densities of primary microglia or neurons were 8000/well with 96 well plates for LDH or CCK-8 test, 1 × 10^5^/dish with confocal dish for AM/PI staining or immunofluorescence staining assay, 2 × 10^5^/well with 12 well plates for q-RT-PCR performing, 5 × 10^5^/well with six well plates for western blot or RNA sequencing.

### 2.2. Administration

Dauricine was purchased from MUST (Chengdu, China, A0315), dissolved in DMSO (Biosharp, BS087, Beijing, China) for mother liquor, and diluted with a medium when treating cells. Limited by the drug administration capacity in mice studies [16,17], dauricine is finally presented as a turbid liquid in 0.9% NaCl saline (XiLong Scientific, Shantou, China). There are many studies on dauricine in transient focal cerebral ischemia in rat and mouse, and the neuroprotective dose is from 5 to 10 mg/kg, so we chose the dosage of 10 mg/kg in this study [17]. The volume of administration was 10 mL/kg for intragastric administration. One dose at the first reperfusion of focal cerebral ischemia was given within 30 min, with one dose given per day. An experiment was conducted on primary microglia cells, in which dauricine or the vehicle were pre-treated for 1 h, followed by LPS (Sigma-Aldricch, 916374, Darmstadt, Germany) (200 ng/mL) to stimulate inflammation, before we collected the cell samples for the next detection.

### 2.3. Oxygen-Glucose Deprivation-Reperfusion

Oxygen-glucose-deprivation-reperfusion (OGD-R) models of primary cortical neurons are well-accepted models of cerebral ischemia reperfusion in vitro [18]. Briefly, the culture mediums for neurons were substituted with Neurobasal^TM^-A medium (1×) liquid that contained no glucose, and a gas mixture composed of 95% N_2_ and 5% CO_2_ was substituted for gas with 5% CO_2_ for 15 min through a hypoxia chamber (Billups-Rothenberg, San Diego, CA, USA). The chamber was sealed and incubated at 37 °C for 15 min, then culture media were substituted to normal medium with glucose and dauricine or DMSO and incubated at 37 °C with 5% CO_2_ for 3 h.

### 2.4. Cell Viability Assays with CCK-8 and LDH

The viability of neurons was assessed by the use of the Cell Counting Kit-8 (CCK-8) product (Dojindo Laboratories, CK04) and a cytotoxicity detection kit (LDH) (Roche, 11644793001, Darmstadt, Germany). We used a post-treatment of OGD and drug for 1 h, then OGD-R for 3 h; next 50 μL of cell supernatant was taken from each well of the 96-well plates, 50 μL LDH working mixture was added away from light, and the cell was added to the new medium with CCK-8 at 37 °C for 1–4 h. The absorbance at 450 nm (CCK-8) and 490 nm (LDH) was measured using a microplate reader. The results obtained for cell viability was presented as the percentage of living cells to control cells.

### 2.5. Calcein Acetoxymethylester/Propidium Iodide AM/PI Staining Assay

The calcein-AM/PI Double Stain Kit (Dojindo Laboratories, C542) was used to detect the viability of neurons. Primary cortical neurons were treated with OGD-R, then incubated with calcein-AM and PI buffer at 37 °C with 5% CO_2_ for 15 min away from light. Next, they were captured with a fluorescence microscope (OLYMPUS, IX73P1F, Tokyo, Janpan). The viable cells were treated with green fluorescence, and dying cells were treated with red fluorescence, so the viability of neurons was counted according to the ratio of green fluorescence to green fluorescence and red fluorescence.

### 2.6. Middle Cerebral Artery Occlusion in Mice

The experimental animal ethics review protocol for Drum Tower Hospital at Nanjing University was approved. The Experimental Animal Ethics Committee examined the protocol, and the studies were conducted according to the guidelines set by Nanjing University’s Guide for Animal Care and Use Committee (approval number: 2021AE01071). All mice experiments were conducted with C57BL/6J (B6) mice (male, eight weeks old; 20 to 24 g body weight) obtained from Nanjing University’s Animal Model Centre. Under anesthesia with isoflurane, the middle cerebral artery was blocked with 6–0 surgical monofilament nylon sutures (Doccol Corporation, Sharon, MA, USA), causing a decrease in blood flow to below 20% [15,19]. The blood flow was restored in the middle cerebral artery (MCA) region followed by occlusion for 1 h. In the sham control mice, the same operation was applied instead of the insertion of the 6–0 surgical monofilament nylon sutures into the MCA. We had a mature and stable technical team, so the mortality was controlled at about 20%, and the success rate of the model was controlled up to 80%.

### 2.7. Infarct Volume Calculation

The mice were euthanized at day 7 after focal cerebral ischemia, and the infarct volume of brain tissue was measured. The experiment was divided into three groups—a Sham group, MCAO group, and a MCAO + Dauricine group with five male mice per group. After cutting the brains coronally into six seria1 1 mm slices, we immersed them in 2% TTC (2,3,5-triphenyltetrazolium chloride, Sigma-Aldricch, T8877, Darmstadt, Germany) for 20 min [20]. ImageJ (the National Institutes of Health, version 1.8.0_172) was used to analyze the images with TTC, and the infarct volume was expressed with the calculation formula percentage lesion volume (%) = (left hemisphere volume-right un-infarcted volume)/(left hemisphere volume × 2) × 100% [21].

### 2.8. Neurological Deficit Scoring

The neurological function of the mice was evaluated until day 7 after MCAO [22], including a rotarod test, forelimb grip strength test, and modified neurological severity score (mNSS) test. The experiment was divided into three groups—a Sham group, MCAO group, and MCAO + Dauricine group with 10 male mice per group. A grip strength meter (BIOSEB, Pinellas Park, FL, USA) was used to measure the forelimb grip strength. In triplicate, the strength of the grip was recorded before release, and the average was recorded at the previous day at the baseline and on days 1, 3, and 7 after MCAO [23]. The rotarod test (IITC Life Science, Woodland Hills, CA, USA) was applied to assess motor deficits and sensorimotor coordination. The mice were pre-trained for three days, as detailed, for the first training session, set at 10/20/30R for 5 min, then adjusted to 20/30/40R for 5 min. It was necessary to immediately replace to the rod to continue training when mice dropped in the process of rotating the rod. Measurements were taken at 40R for 5 min on the previous day at the baseline and on days 1, 3, and 7 after MCAO [24]. The mNSS test was used to measure sensory function, motor function, reflexes, and balance, and was administered on days 1, 3, and 7 after MCAO [25]. During the rating of the mice, the investigator was blinded to the experimental groups.

### 2.9. Cerebral Blood Flow Measure

Utilizing the PeriFlux System 5000 (Perimed AB, Järfälla, Sweden), the cerebral blood flow was strictly monitored during the MCAO modeling, the model success was verified, and injury was assessed from the cerebral blood flow measured using a PeriCam PSI (Perimed AB, Järfälla, Sweden) post-operation.

### 2.10. RNA Isolation and Quantitative Real-Time PCR

Tissue and cell samples were lysed and extracted with the RNA Isolation Kit (FastPure Cell/Tissue Total RNA Isolation Kit V2, Vazyme, RC112). Quantitative real-time PCR was performed in a Roche LightCycler^®^ 96 system (Roche, Mannheim, Germany) with a SYBR green kit (Accurate, China, AG11701). The primers used were as shown in Table 1:

### 2.11. RNA Sequencing

The sequencing was performed by Majorbio (Shanghai, China), based on the Illumina Novaseq 6000 Sequencing platform. The percentage of Q30 base was above 95.73%, and the amount of clean data from each sample were above 6.35 Gb. Based on the quantitative results obtained for the expression levels, the differential genes between groups were analyzed using differential Venn analysis, Reactome enrichment analysis, functional enrichment analysis, cluster analysis, and KEGG enrichment analysis. Each group had three duplicate samples.

### 2.12. Cytokine Analyses in Cellular Supernatant

We used the Luminex liquid suspension chip from Wayen Biotechnologies (Shanghai, China), and monitored the concentration of cytokine in cellular supernatant with the Bio-Plex Mouse Panel 23-plex Cytokine kit [26]. In brief, 50 μL samples or standard substance were added to 96-well plates and incubated for 30 min in room temperature away from light; then, we added 25 μL of diluted detection antibody and incubated the mixture for 30 min as before. Next, we added 50 μL of diluted streptavidin-PE and incubated the mixture for 10 min. Finally, the signal was measured using the Bio-Plex 200 System (Luminex Corporation, Austin, TX, USA).

### 2.13. Western Blot Assay

The RIPA lysis buffer (Thermo Scientific, Waltham, MA, USA, 89901) was used to extract the total protein. In brief, 10% SDS-PAGE was adopted to separate the protein and the protein was electrophoretically transferred onto 0.2 μm PVDF membranes (Immobilon-PSQ Transfer membrane, Merck Millipore, ISEQ00010). Then 5% skim milk was applied to block the membranes for 2 h at room temperature; then, we incubated them at 4 °C for at least 12 h with the following antibodies: anti-Stat5 antibody (CST, 9363S, 1:1000), anti-phospho-Stat5 antibody (CST, 4322S, 1:1000) (endogenous levels of Stat5a only when phosphorylated with Tyr694 and endogenous levels of Stat5b when phosphorylated with Tyr699), anti-P-NF-kB antibody (CST, 3033S, 1:1000), anti-NF-κB antibody (CST, 8242S, 1:1000), anti-β-tubulin antibody (Bioworld Technology, AP0064, 1:2000), and anti-β-actin antibody (Bioworld Technology, AP0060, 1:2000). Corresponding secondary antibodies were chosen for combination with the membranes for 1 h at room temperature and were visualized with the Chemiluminescent HRP Substrate (Millipore, Burlington, MA, USA, WBKLS0500). ImageJ (the National Institutes of Health, version 1.8.0_172) was used to quantify the bands with a gray value. The relative protein expression was expressed with the band intensity for the β-tubulin or β-actin.

### 2.14. Immunofluorescence Staining in Brain Sections and Microglia Cells

Mice were perfused with PBS and 4% paraformaldehyde through the left ventricle while in deep anesthesia, followed dehydrated with glucose solution, then sectioned with a 20 μm thickness. Primary microglia cells were treated with lipopolysaccharide or dauricine for 3 h, then we discarded the supernatant. The brain slices or microglia cells were processed using the same operation in the next stain, fixed in 4% paraformaldehyde for 15 min, and 0.25% Triton X-100 was used to permeabilize them for 15 min. After this, 2% bovine serum albumin was applied to block them for 2 h, followed by incubating them with indicated anti-Iba1 antibody (Abcam, ab5076, 1:100) at 4 °C for at least 12 h. The next day, the antibody was recycled, and the slices and cells were gently washed in PBS three times; then, the corresponding secondary antibodies with a 594 nm wavelength (Invitrogen, 1:500) were incubated for 2 h, avoiding light, at room temperature. The DAPI staining kit (Bio-world, BD5010, 1:1000) was used to counterstain the nuclei. A fluorescence microscope (OLYMPUS, IX73P1F, Tokyo, Japan) and confocal laser scanning microscope (OLYMPUS, FV3000, Tokyo, Janpan) were applied to obtain the images, and a positive amount was quantified with ImageJ (the National Institutes of Health, version 1.8.0_172).

### 2.15. Ligand–Receptor Docking and Analysis

The AutoDock 4 program (version 4.2.6) was used to perform the Ligand–receptor docking. The binding site was analyzed, and images were generated with PyMOL, version 2.2.0 [27]. The STAT5a and STAT5b 3D protein structure were derived from PDB (Protein Data Bank, www.rcsb.org (accessed on 15 February 2022). The dauricine 3D structure was taken from PubChem (pubchem.ncbi.nlm.nih.gov, accessed on 20 June 2022).

### 2.16. Plasmid Construction and Transfection

We acquired the STAT5b transcription (NM_001113563) from NCBI https://www.ncbi.nlm.nih.gov (accessed on 20 February 2022) and designed the STAT5b overexpression vector (pLV11ltr-Puro-mCherry-CMV-STAT5b) (Abbr: STAT5b OV) and STAT5b mutant with Tyr-699 (pLV11ltr-Puro-mCherry-CMV-STAT5b(Y699A)) (Abbr: STAT5b (Y699A)). Using a point mutation primer, PCR point mutation was used to mutate the 699th amino acid Tyr (Tyrosine) to Ala (Alanine). The plasmid construction work was mainly carried out by Nanjing Corues Biological Co., Ltd. (Cornes Biological, Nanjing, China). After a sequencing reaction, the target sequence was verified to be correctly connected to the target vector. Plasmid transfection was performed according to the instructions of Lipofectamine^TM^ 3000 (Invitrogen, L3000015); 48 h later, the expression of red fluorescence (mCherry) was observed under a fluorescence microscope, and the fluorescence expression rate directly reflected the cell transfection rate.

### 2.17. Statistical Analysis

All values are expressed as the mean ± SEM. An unpaired Student’s *t*-test was used for two-group comparisons, and two-way ANOVA followed by Dunnett’s test was used to analyze the quantitative variables for multiple pairwise comparisons. Statistically significance was defined as *p* < 0.05. GraphPad Prism 8 (version 8.0.2, GraphPad Software, San Diego, CA, USA) was used to create the statistical graph.

## 3. Results

### 3.1. Dauricine Decreased Neuron Death Induced by OGD-R

First, we examined the potential neurotoxicity of dauricine on primary neurons through adding different concentrations of dauricine to the media for 3 h; then, we determined the LDH of culture supernatant and the CCK8 of cells. The compound structure of dauricine was as presented in Figure 1A. Dauricine did not decrease the cell viability and showed no significant cytotoxicity for primary neurons below 40 μM (Figure 1B,C), and the results of LDH and CCK8 showed the same trend. Thus, the cell viability of OGD-R-treated neurons was detected using the LDH of the cell supernatant, and we found a significant increase caused by dauricine (1 μM) (*p* < 0.01; Figure 1F). Furthermore, dauricine treatment was found to partially rescue the neuronal cell death induced by OGD-R, as shown by the calcein-AM and PI staining (*p* < 0.05; Figure 1D,E).

### 3.2. Dauricine Decreased Brain Infarct Size and Neurological Deficits after tMCAO Injury

We verified whether the administration of dauricine has the same neuroprotective in vivo as in vitro. An infarction of the right hemisphere was developed after 1 h of MCAO, followed by reperfusion after 1 day, 3 days, and 7 days, and the mouse studies were conducted through behavior experiments (Figure 2A). From the cerebral blood flow after MCAO, it was seen that the blood flow of the infarct side brain significantly dropped, showing the success of model (Figure 2B). The TTC staining method was used to evaluate the cerebral infarct volume. Compared to the vehicle-treated group, the administration of dauricine showed a 56% reduction in infarct volume at 7 day after MCAO (Figure 2C,D; *p* < 0.05). As shown in Figure 2E–G, the group treated with dauricine had significantly better neurological function at day 3 and day 7 after MCAO than the vehicle-treated group (*p* < 0.05).

### 3.3. Dauricine Inhibited the Activation of Microglia In Vitro and In Vivo

Iba1 is a marker of microglia cell activation, and microglia displays as an amebocyte when it is activated, so the cell area also partly represents the activation of microglia cells. It was found that dauricine inhibited the activation of microglia from injury brain in three-dimensional and plan views (Figure 3A). Then, dauricine was also found to reduce the activation and cell area of amoeboid microglia cells induced by LPS in vitro (Figure 3B–D).

### 3.4. Dauricine Reduced Inflammation Induced by LPS or tMCAO Injury

Inflammation plays a causal role in the pathogenesis of stroke, a fact that is supported by many sources of evidence. We found that the levels of inflammation biomarkers IL-1β, IL-6, and TNF-α were increased by MCAO and reduced by dauricine in the ischemic penumbra of tMCAO on day 7 (Figure 4A). Then, the anti-inflammation effect of dauricine in the primary microglia induced by LPS was verified. The primary microglia samples collected were subjected to dauricine (1 μM) pretreatment for 1 h and LPS or dauricine stimulation for another 3 h; then, the expression of inflammation factors expression was detected using q-RT-PCR or Luminex liquid suspension chip (Figure 4B). We found that the levels of inflammation biomarkers IL-1β, IL-6, and TNF-α were significantly increased by LPS and reduced by dauricine in vitro (Figure 4C).

Inflammation and immune cytokines communicate and work with cells mainly through proteins, so we measured the inflammation cytokines using a Luminex liquid suspension chip for the cell supernatant. With the LPS- induced inflammation storm, dauricine significantly reduced the cytokines as follows (Figure 4D): Eotaxin, KC, TNF-α, IL-1α, IL-1β, IL-6, IL-12β, and IL-17α.

### 3.5. Dauricine Down-Regulated Expression of Immune Factors

We applied RNA sequencing to further explore the mechanism by which dauricine reduces inflammation. The distribution of reads compared to the genome in different regions of the reference genome was statistically analyzed and the percent of reads mapped to genome regions was shown, demonstrating that CDS made up 71.95%. Compared with the LPS group, the dauricine group had 542 upregulated genes and 313 downregulated genes (pad-just ≤ 0.05, fold ≥ 1.5) (Figure 5A). Venn analysis shown that 188 genes were common among both the Control vs. LPS and LPS vs. LPS + Dauricine gene sets (Figure 5B). The reactome enrichment analysis revealed that the main reactome pathways were G alpha (q) signaling events and cytokine signaling in the immune system in LPS vs. LPS + Dauricine. Therefore, we focused on the common 188 genes and immune system downregulated sets, Venn analysis also confirmed that 14/15 genes of the immune system that were downregulated were included in the common set of 188 genes (Figure 5C). The functional enrichment analysis showed the enrichment of the 14 gene sets, and the hypergeometric distribution algorithm was used to obtain the significant enrichment functions of genes in the gene set. The heatmap of the immune system downregulated set showed the expression of 15 genes among the Control, LPS, and LPS + Dauricine groups (Figure 5D). To further explore the pathway of immune system downregulation, the KEGG enrichment analysis was applied and shown the JAK-STAT, TNF, and NF-κB signaling pathways were showed to be involved (Figure 5E).

### 3.6. Dauricine Inhibited the Activation of STAT5 and NF-κB Pathways

The JAK-STAT and NF-κB signaling pathways have been singled out by RNA sequencing, which might be involved in the regulation of inflammation regulated by dauricine. Therefore, we focused on the protein expression and modification of the JAK-STAT and NF-κB signaling pathways. We found that dauricine significantly inhibited the P-STAT5 and P-NF-κB induced by LPS in microglia cells (Figure 6A–C), which indicated that dauricine might be able to reduce inflammation through regulating the STAT5-NF-κB signaling pathway.

Ligand–receptor docking was performed with AutoDock and analyzed with PyMOL. The molecular docking energy (<−5 kj/mol) was sufficiently low, which demonstrated that dauricine could directly bind with STAT5b at Tyr-699, the main phosphorylation region (Figure 6E), but the binding force with STAT5a was not promising (Figure 6D). This indicated that dauricine might target STAT5b to induce effects.

### 3.7. STAT5b (Tyr699 mutant) Reversed the Neuroprotection of Neuron and Activated the NF-κB Pathway of Microglia

To further clarify the neuroprotective and anti-inflammatory effects of dauricine including directly targeting STAT5 and inhibiting STAT5b phosphorylation, we constructed a STAT5b overexpression plasmid (STAT5b OV) and a STAT5b phosphorylation site Tyr-699 mutation-STAT5b (Y699A) plasmid (Figure 7A,B) to intervene neurons. The results showed that STAT5b (Y699A) reversed the protective effect of dauricine on the oxygen-glucose deprivation-reperfusion injury of neurons (Figure 7C,D). The results obtained suggest that dauricine might be able to directly bind to the Tyr-699 phosphorylation region of STAT5b to achieve its neuroprotective effect.

We also detected the effect of STAT5b (Y699A) on the NF-κB pathway and found that STAT5b (Y699A) reversed the suppression of dauricine on the P-NF-κB of microglia (Figure 7E,F). These results suggested that dauricine might bind to the Tyr-699 phosphorylation region of STAT5b to suppress the activation of the NF-κB pathway.

## 4. Discussion

Plaque inflammation contributes to stroke, and the identification of recurrent stroke is improved by the risk score of carotid plaque inflammation [5]. Inflammatory cascade starts in hypo-perfused vessels and the ischemic brain parenchyma, and inflammatory mediators generate the systemic immune response [28]. In the acute phase of ischemic stroke, first microglia are activated, and then a large amount of inflammation occurs and chemokines are released. After this, other immune cells invade to the peri-infarct and infarct core, and finally the inflammatory response is aggravated by the resident and infiltrating immune cells together, including cytokines [29]. The dynamic balance between pro- and anti-inflammatory responses can be better understood as a basis for designing effective therapies [30]. We focused on the mechanism of inflammation and stroke and found that the new inhibitor of STAT5 has a good anti-inflammation effect and neuroprotective effect in the in vivo and in vitro experimental models of stroke. Dauricine has been recognized as an inhibitor of STAT5 in this study for perhaps first time. It exhibited a powerful anti-inflammatory effect against the inflammation storm induced by LPS in the primary microglia and reduced the brain injury in the MCAO-reperfusion mouse model.

The STAT5 and related signaling pathways play a crucial role in autoimmunity and neuroinflammation [31,32]. The STAT5 deficiency induced a lack of γδT17 cells and led to a profound resistance to experimental autoimmune encephalomyelitis. The γδT17 cell expansion was promoted and gut-associated T-bet was mainly downregulated by STAT5a homolog rather than STAT5b [31]. The STAT5 tetramers were showed to promote the pathogenesis of experimental autoimmune encephalomyelitis, and the production of CCL17 was regulated by GM-CSF-mediated STAT5 tetramerization through monocyte-derived cells in the STAT5 tetramer-deficient N-domain double knockout mouse [32]. The Th17 cells were converted into cells that mediate IL-9-dependent effects by STAT5 and BATF in allergic airway inflammation and anti-tumor immunity [33]. The threshold of STAT3, STAT5a, and STAT5b expression was showed to determine whether the levels of PRR-induced proinflammatory cytokines need to be increased or decreased [34]. The STAT5 inhibitor substantially decreased inflammation in cardiac hypertrophy in Ang II-induced mice [35], and the STAT5 inhibitor significantly attenuated atherosclerosis via decreasing inflammation in ApoE^−/−^ mice induced by HFD [36]. Therefore, STAT5 is an important therapeutic target in anti-inflammation, and the inhibitor has great pharmacological value. In our study, we found a new inhibitor of STAT5, dauricine, which regulates the function of microglia, plays a powerful role in anti-inflammation, and protects the brain against neuroinflammation post-MCAO reperfusion. The tools AutoDock and PyMOL found that dauricine binds with STAT5b at the phosphorylated region of Tyr-699 more strongly than STAT5a.

The NF-κB plays a vital role in the expression of proinflammatory genes including cytokines, chemokines, and adhesion molecules; therefore, is widely considered as a proinflammatory signaling pathway, furthermore as a target for new anti-inflammatory drugs, being called the “holy grail” by some [37]. The NF-κB is involved in acute inflammation induced by LPS [38] and carrageenan [39], as well as chronic inflammation in conditions such as type 2 diabetes [40], oral mucosa inflammation [41], environment-derived osteoarthritis [42], etc. The STAT5 and NF-κB have been assessed to be present in statistically higher amount in the peripheral blood leukocytes of patients with axSpA [43] The suppression of NF-κB can mediate inflammation through the JAK2-STAT5 pathway [44]. Therefore, we propose that dauricine reduced the neuroinflammation of post-MCAO reperfusion via mediating NF-κB activation by inhibiting the STAT5.

In focal cerebral ischaemia and reperfusion in rats, STAT5b was confirmed to be significantly modulated in the hippocampus [45]. However, the research on the use of STAT5b in brain injury caused by focal cerebral ischemia and reperfusion has not yet been widely accepted, and its relation to neuroinflammation has not been -studied in detail. In our study, we focused on stroke with neuroinflammation, and revealed the effect of STAT5 on stroke through anti-inflammation, as shown in Figure 8. Dauricine was first to be suggested as a new inhibitor of STAT5, and its effect on the potential therapeutic target of STAT5b inhibitor on the neuroinflammation present in brain injury related to MCAO-reperfusion was exhibited; however, more evidence, such as on its effect on STAT5b knockout mice, is needed to clarify this hypothesis in future works.

## 5. Conclusions

In this study, dauricine was shown to suppress the inflammatory storm and protect injury from post-ischemia-reperfusion. This is the first study to find that dauricine regulates microglia activation and protects neurons from ischemia-reperfusion injury possibly through inhibiting the STAT5- NF-κB pathway. These findings have revealed that the anti-inflammatory and neuroprotective mechanisms of dauricine might lead to the suppression of microglia activation via targeting the inhibition of STAT5-NF-κB in ischemic stroke.

## Figures and Tables

**Figure 1 brainsci-12-01153-f001:**
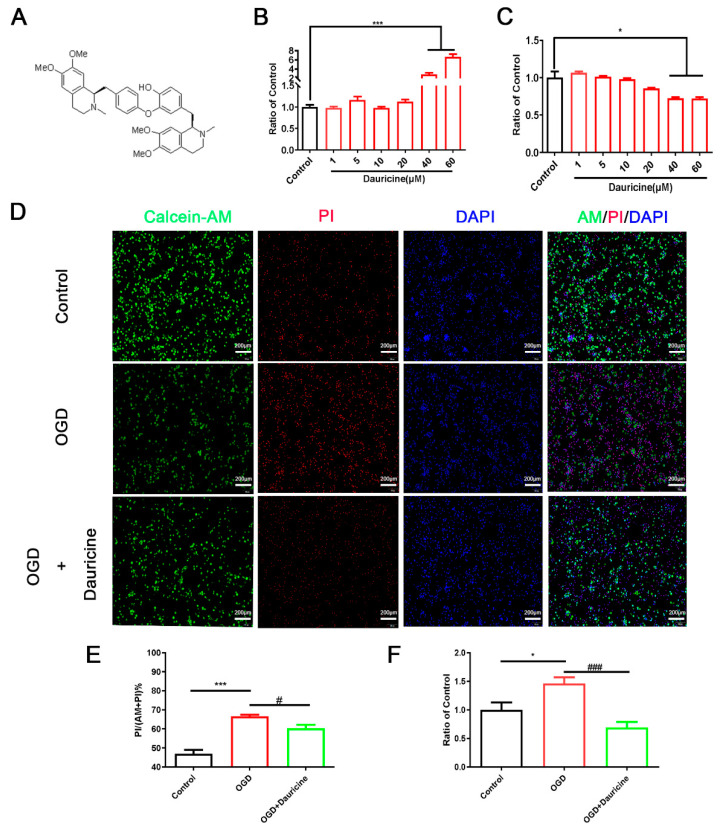
Dauricine decreased the primary neuron death induced by OGD-R. (**A**) The compound structure of dauricine. (**B**) LDH of primary cortical neurons cell supernatant, and (**C**) CCK8 of primary cortical neurons cells were detected following dauricine (1, 5, 10, 20, 40, 60 μM) treatment for 24 h, *n* = 5. Control group vs. Dauricine group, * *p* < 0.05, *** *p* < 0.001. Oxygen-glucose was applied to primary cortical neuron cells for 15 min; then, they were reperfused and treated with dauricine (1 μM) for 3 h. (**F**) The viability of primary cortical neuron cells was detected by LDH, and cell viability was also showed by calcein-AM and PI buffer (**D**) and quantified (**E**), *n* = 5. Control group vs. OGD group, * *p* < 0.05, *** *p* < 0.001, OGD group vs. (OGD + Dauricine) group # *p* < 0.05, ### *p* < 0.001, unpaired Student’s *t*-test.

**Figure 2 brainsci-12-01153-f002:**
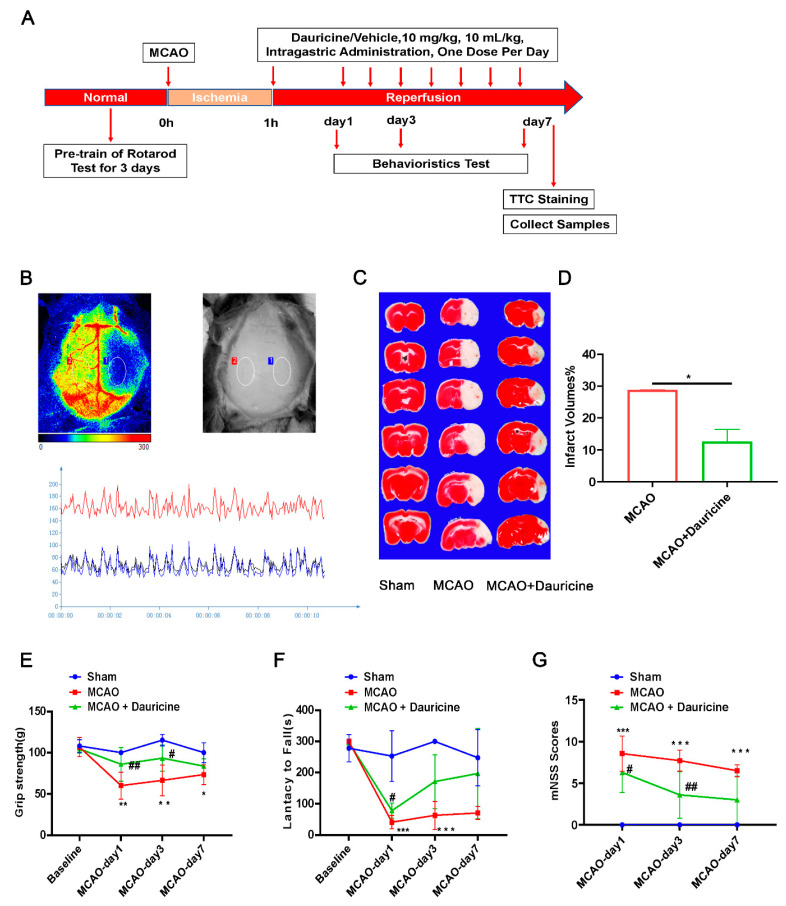
Dauricine improved the tMCAO injury. (**A**) The diagram of the animal studies with the experimental design timeline. (**B**) The cerebral blood flow after MCAO, anatomy and infrared images are showed above, while the quantitative figure is showed below. (**C**) Mouse brains were procured at day 7 after ischemia-reperfusion, and the infarct volume was determined by TTC staining and quantified (**D**) (* *p* < 0.05). The functional outcomes of the tMCAO mice were evaluated by grip strength (**E**), Lantacy to fall (**F**), and mNSS scores (**G**). *n* = 10 mice per group, Sham group vs. MCAO group, ** *p* < 0.01, *** *p* < 0.001, MCAO group vs. (MCAO + Dauricine) group # *p* < 0.05, ## *p* < 0.01, two-way ANOVA with Bonferroni post hoc test.

**Figure 3 brainsci-12-01153-f003:**
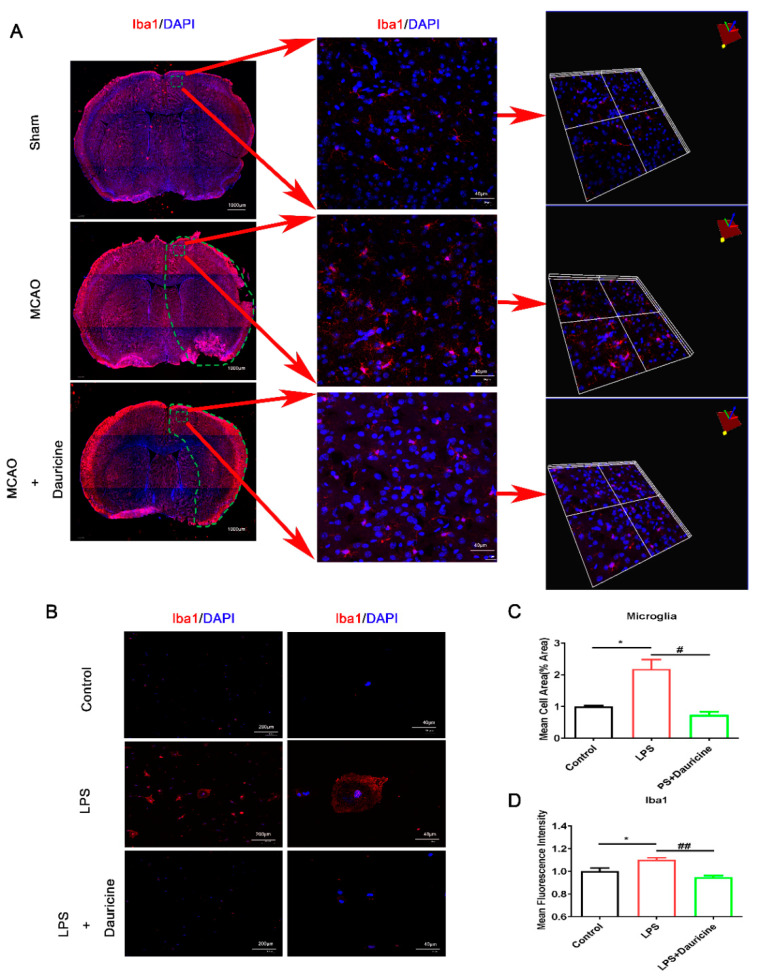
Dauricine inhibited the activation of microglia in vitro and in vivo. (**A**) The representative figures of brain sections in MCAO at day 7 strained with immunofluorescence of Iba1 and DAPI are shown in the full brain and in a three-dimensional view. (**B**) The representative figures of primary microglia cells induced by LPS or dauricine for 3 h in vitro. The mean cell area (%) of microglia cells was counted (**C**) and the mean fluorescence intensity of Ibal was counted (**D**). *n* = 3, Control group vs. LPS group, * *p* < 0.05; LPS group vs. (LPS + Dauricine) group, # *p* < 0.05, ## *p* < 0.01, unpaired Student’s *t*-test.

**Figure 4 brainsci-12-01153-f004:**
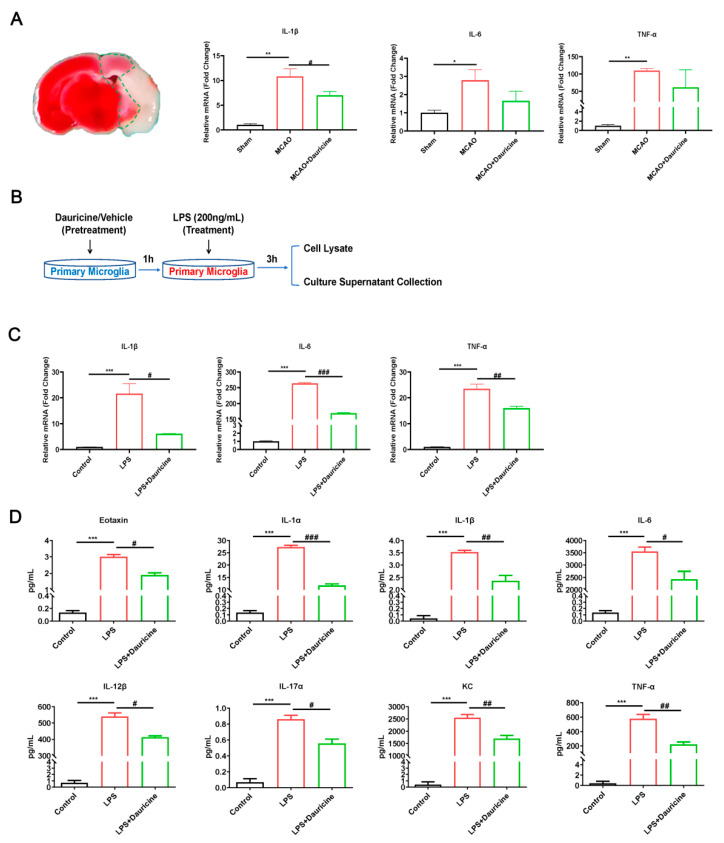
Dauricine decreased inflammation induced by LPS or tMCAO injury. (**A**) Diagram of ischemic penumbra, and samples from homogenate of brain infarct were analyzed with the inflammation mRNA IL-1β, IL-6 and TNF-α at tMCAO day 7, *n* = 3. Sham group vs. MCAO group, * *p* < 0.05, ** *p* < 0.01; MCAO group vs. MCAO + Dauricine group, # *p* < 0.05. (**B**) The diagram of LPS or dauricine stimulated primary microglia, and collected the samples as shown. (**C**) The microglia cell lysate was measured with inflammation mRNA IL-1β, IL-6 and TNF-α, *n* = 3. Control group vs. LPS group, *** *p* < 0.001; LPS group vs. (LPS + Dauricine) group, # *p* < 0.05, ## *p* < 0.01, ### *p* < 0.001, unpaired Student’s *t*-test. (**D**) The inflammation cytokines measured by Luminex liquid suspension chip for the cell supernatant, and cytokines as follows: Eotaxin, IL-1α, IL-1β, IL-6, IL-12β, IL-17α, KC, TNF-αn = 3, Control group vs. LPS group, *** *p* < 0.001; LPS group vs. (LPS + Dauricine) group, # *p* < 0.05, ## *p* < 0.01, ### *p* < 0.001, unpaired Student’s *t* test.

**Figure 5 brainsci-12-01153-f005:**
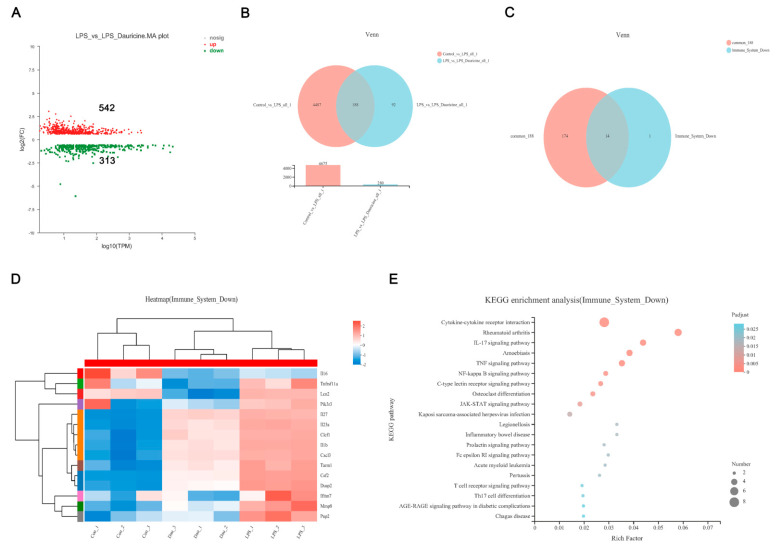
Transcriptomics of LPS or dauricine stimulated primary microglia. (**A**) MA diagram of the expression difference between the LPS and LPS + Dauricine groups. (**B**) Venn analysis of the difference between the Control vs. LPS and LPS vs. LPS + Dauricine sets. (**C**) Venn analysis of the difference between the common 188 genes and immune system downregulated sets. (**D**) Heatmap of the immune system downregulated set and KEGG enrichment analysis of the immune system downregulated set of the 14 gene sets from the common portion of the common 188 genes and immune system downregulated sets (**E**).

**Figure 6 brainsci-12-01153-f006:**
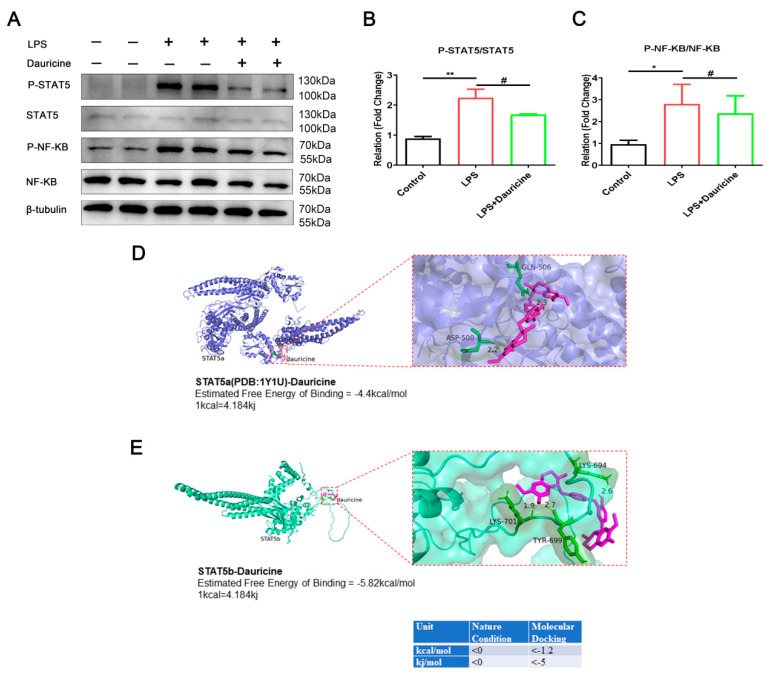
Dauricine inhibited the activation of STAT5, NF-κB and docked with STAT5. (**A**) The western blot representative figure of NF-κB, P-NF-κB, STAT5, P-STAT5 and β-tubulin were showed and was counted with ratio of P-STAT5 to STAT5 (**B**) and P-NF-κB to NF-κB (**C**). *n* = 3, Control group vs. LPS group, * *p* < 0.05, ** *p* < 0.01; LPS group vs. (LPS + Dauricine) group, # *p* < 0.05, unpaired Student’s *t*-test. Ligand receptor docking was performed with AutoDock and analyzed with PyMOL. The top panel shows the joint diagram of dauricine and STAT5a protein (**D**), and the lower panel shows the joint diagram of dauricine and STAT5b protein (**E**), the energy data under natural conditions and molecular docking is noted at bottom right.

**Figure 7 brainsci-12-01153-f007:**
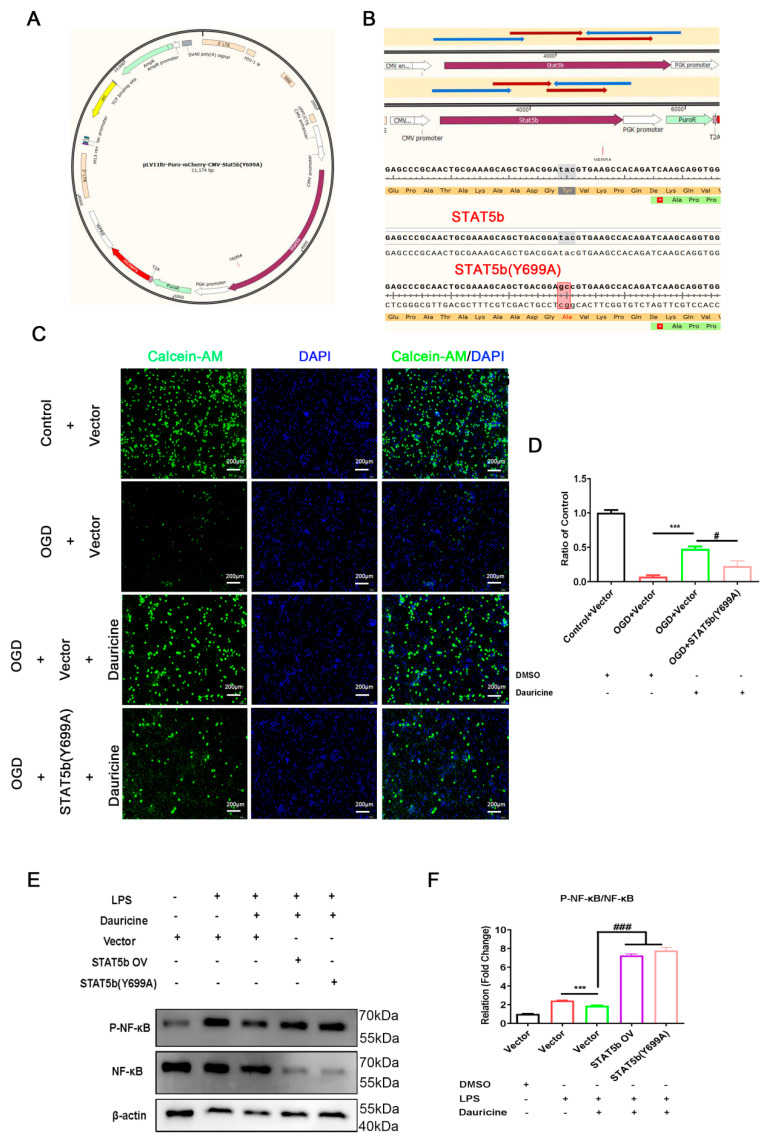
STAT5b (Tyr699 mutant) reversed the neuroprotection of neuron and activated the NF-κB pathway of microglia. (**A**) Vector map of STAT5b (Tyr-699 mutation). (**B**) Sequencing and matching results of STAT5b OV and STAT5b (Y699A). (**C**) The primary neurons were transfected with STAT5b OV and STAT5b (Y699A) plasmids for 48 h, and the neurons were deprived of oxygen and sugar for 30 min, then reperfusion for 3 h, and neurons were stained with calcein-AM and DAPI, and quantified (**D**). *n* = 5, (OGD + Vector) group vs. (OGD + Vector +dauricine) group, *** *p* < 0.001; (OGD + Vector + dauricine) group vs. (OGD + STAT5b (Y699A) + dauricine) group, # *p* < 0.05, unpaired Student’s *t*-test. (**E**) STAT5b OV and STAT5b (Y699A) plasmids were transfected into primary microglia for 48 h, and induced by LPS or LPS + dauricine for 3 h, then detected by the western blot and representative figure of NF-κB, P-NF-κB, and β-actin were shown, and was counted with ratio of P-NF-κB to NF-κB (**F**). *n* = 3, (Vector + LPS) group vs. (Vector + LPS + Dauricine) group, *** *p* < 0.001; (Vector + LPS + Dauricine) group vs. (STAT5b OV + LPS + Dauricine) group or (STAT5b (Y699A) + LPS + Dauricine) group, ### *p* < 0.001, unpaired Student’s *t*-test.

**Figure 8 brainsci-12-01153-f008:**
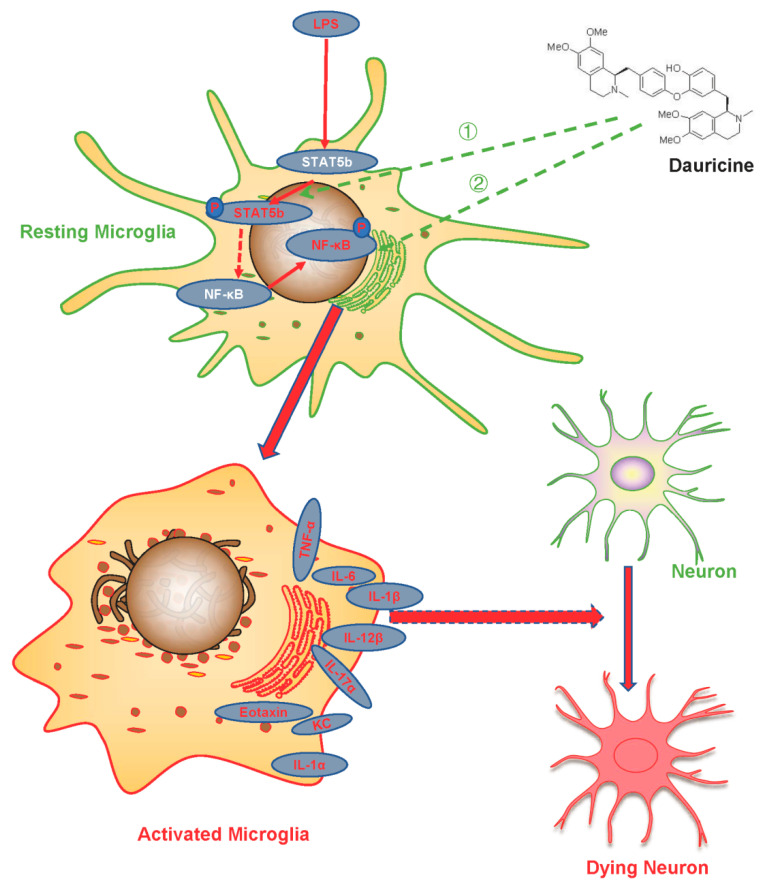
Schematic diagram of the mechanisms underlying the role of dauricine on regulating the microglia activation and protecting neurons from ischemia-reperfusion injury.

**Table 1 brainsci-12-01153-t001:** Primers Information Table.

Gene	Forward Primer (5′ to 3′)	Reverse Primer (5′ to 3′)
TNF-α	CCTGTAGCCCACGTCGTAG	GGGAGTAGACAAGGTACAACCC
IL-1β	GAAATGCCACCTTTTGACAGTG	TGGATGCTCTCATCAGGACAG
IL-6	TAGTCCTTCCTACCCCAATTTCC	TTGGTCCTTAGCCACTCCTTC
GAPDH	AGGTCGGTGTGAACGGATTTG	TGTAGACCATGTAGTTGAGGTCA
β-Actin	GGCTGTATTCCCCTCCATCG	CCAGTTGGTAACAATGCCATGT

## Data Availability

Data sharing not applicable to this article as no datasets were generated or analyzed during the current study.

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
