# Peer review of "Regulation of Microglia-Activation-Mediated Neuroinflammation to Ameliorate Ischemia-Reperfusion Injury via the STAT5-NF-κB Pathway in Ischemic Stroke"

_brainsci, 2022, doi:10.3390/brainsci12091153_

Round 1
Reviewer 1 Report
Review for (Brain Sciences) Manuscript ID - brainsci-1876198
The manuscript by Zhijun Pu et al., entitled "Regulation microglia activation mediated neuroinflammation to ameliorate ischemia-reperfusion injury via STAT5-NF-κB pathway in ischemic stroke" is the research report finding that the anti-inflammation of dauricine is mainly through STAT5-NF-κB pathway, might act as a STAT5 inhibitor. Dauricine suppressed the inflammation cytokines Eotaxin, KC, TNF-a, IL-1a, IL-1ß, IL-6, IL-12ß, IL-17a, and also inhibited the microglia activation. STAT5b mutant at Tyr-699 reversed the protective effect of dauricine on the oxygen-glucose deprivation-reperfusion injury of neurons, and reactivated the suppression of dauricine on P-NF-κB of microglia. These results suggest that dauricine suppresses neuroinflammation and protects the neuron from the injury of post-ischemia reperfusion via mediating the microglia activation through the STAT5-NF-κB pathway, and a potential therapeutic target for neuroinflammation. The manuscript is well written the results are well presented. However, some remarks should be taken by authors under consideration before paper publication.
Comments:
1. Authors may represent the animal studies with the experimental design timeline in the figure will be useful and easily understandable to the readers.
2. Authors should explain the mice for the rotarod test were pre-trained or not for the test.
3. Authors should include the details procedure for the rotarod test and grip strength measurement test.
4. Authors should include the table for real-time PCR forward and reverse primers. It will be reader-friendly.
5. Authors should include the molecular weight of each western blot image (Fig 6 & 7).
6. In immunohistochemistry data representation authors should change the group name from the “horizontal” scale to the “vertical” scale.
Author Response
Dear editors and reviewers:
We much appreciated the valuable time of editors and reviewers in reviewing our manuscript, especially for the speed of dealing. Those comments are very professional and helpful for revising and improving our paper. According to the instructions provided, we have thoroughly revised the manuscript including figures and writing. After revision, the manuscript has again further been sent to the MDPI for language editing services (Invoice ID: English-49234). We have tried our best to revise every point of the manuscript according to the suggestions from authoritative peers and make some appropriate supplementary data. I will beg for the speed of dealing again, because my Ph.D. student wants to get his degree this September, and this article is very important and urgent for his degree and future job. Instead, we have explained and made it clear point-by-point to the reviewers’ comments as followed. (Revised manuscript with tracked in the attachment)
- Authors may represent the animal studies with the experimental design timeline in the figure will be useful and easily understandable to the readers.
Response: Thanks for the good suggestion, and we added the diagram of the animal studies with the experimental design timeline to Figure 2A in the revised manuscript.
- Authors should explain the mice for the rotarod test were pre-trained or not for the test.
Response: Thanks for the carefully review. It`s due to the limited space and thought it`s a mature and well-known method, we didn`t describe the details, but we tested the behavior experiments fully according to the related standard operation and the rotarod test were surely pre-trained for three days. Pre-train as be detailed, for the first training session, set (10/20/30R, 5 mins), then adjust to (20/30/40R, 5 mins), it is necessary to immediately re-place to the rod to continue training when mouse drop in the process of rotating the rod. We have supplemented the detailed description of the behavior experiments in the revised manuscript.
- Authors should include the details procedure for the rotarod test and grip strength measurement test.
Response: Thanks for the suggestion, and we have supplemented the detailed description of the behavior experiments including the rotarod test and grip strength measurement in the revised manuscript.
- Authors should include the table for real-time PCR forward and reverse primers. It will be reader-friendly.
Response: Thanks for the good suggestion, and we have added the table of real-time PCR forward and reverse primers in the appropriate position.
- Authors should include the molecular weight of each western blot image (Fig 6 & 7).
Response: Thanks for your suggestion. We have re-edited and included the molecular weight of each western blot image in Fig 6 & Fig 7.
- In immunohistochemistry data representation authors should change the group name from the“horizontal” scale to the “vertical” scale.
Response: Thanks for your criticism, we have changed the group name from the “horizontal” scale to the “vertical” scale in all the immunohistochemistry representation figures.

Reviewer 2 Report
The manuscript by Pu et al. demonstrated that a Chinese traditional medicine, dauricine, reduces neural damage in models of brain injury both in vitro and in vivo. The authors also showed that dauricine protects neurons by preventing microglia activation, through its inhibitory effects on STAT5-NF-κB pathway in microglia. The experiments were well designed and carried out, the conclusions drawn by the authors were largely supported by their results. However, there are major concerns over manuscript writing.
1. The authors need to greatly improve their English writing, there are numerous grammatical errors throughout the manuscript. Please send the manuscript to others for proofreading before next submission.
2. Many sentences are hard to understand due to ambiguity.
For example, in the Abstract the authors wrote “STAT5 is proving to be a highly effective anti-inflammatory therapy with great potential, and inhibition of STAT5 has demonstrated significant anti-inflammation and therapeutic effects, but rarely focus on mechanism of neuroinflammation and brain injury from ischemia-reperfusion”. The first two part2 of the sentence seem to be redundant, and the third part is hard to understand. Do you mean ”Targeting STAT5 has been shown to be a highly effective anti-inflammatory therapy, however, the mechanism by which STAT5 signaling pathway regulate neuroinflammation following brain injury induced by ischemia-reperfusion remains unclear”?
The authors need to fix many ambiguous sentences like this one throughout the manuscript. Try to shorten the sentences and make your statement clear in each sentence.
3. The authors needs to improve their Abstract. More specifically, please state the major unknown questions you are trying to solve, and provide sufficient information on dauricine and why you studied it.
Minor points:
1. In the introduction, the authors claim”Microglia is the only resident immunoreactive cells in the brain”. This statement is not true, perivascular macrophages are also resident immune cells.
2. Figure 5 is hard to read. The authors may consider removing some panels (such as panel A) and make fronts on the others clearer.
Author Response
Dear editors and reviewers:
We much appreciated the valuable time of editors and reviewers in reviewing our manuscript, especially for the speed of dealing. Those comments are very professional and helpful for revising and improving our paper. According to the instructions provided, we have thoroughly revised the manuscript including figures and writing. After revision, the manuscript has again further been sent to the MDPI for language editing services (Invoice ID: English-49234). We have tried our best to revise every point of the manuscript according to the suggestions from authoritative peers and make some appropriate supplementary data. I will beg for the speed of dealing again, because my Ph.D. student wants to get his degree this September, and this article is very important and urgent for his degree and future job. Instead, we have explained and made it clear point-by-point to the reviewers’ comments as followed. (Revised manuscript with tracked in the attachment)
- The authors need to greatly improve their English writing, there are numerous grammatical errors throughout the manuscript. Please send the manuscript to others for proof reading before next submission.
Response: Thanks for the criticism, and the manuscript has again further been sent to the MDPI for language editing services (Invoice ID: English-49234) before this submission.
- Many sentences are hard to understand due to ambiguity. For example, in the Abstract the authors wrote “STAT5 is proving to be a highly effective anti-inflammatory therapy with great potential, and inhibition of STAT5 has demonstrated significant anti-inflammation and therapeutic effects, but rarely focus on mechanism of neuroinflammation and brain injury from ischemia-reperfusion”. The first two part 2 of the sentence seem to be redundant, and the third part is hard to understand. Do you mean “Targeting STAT5 has been shown to be a highly effective anti-inflammatory therapy, however, the mechanism by which STAT5 signaling pathway regulate neuroinflammation following brain injury induced by ischemia-reperfusion remains unclear”? The authors need to fix many ambiguous sentences like this one throughout the manuscript. Try to shorten the sentences and make your statement clear in each sentence.
Response: Thanks for the carefully read and professional criticism, we have thoroughly revised the manuscript twice again and try to shorten the sentences and make your statement clear in each sentence. After revision, the manuscript has further been sent to the MDPI for language editing services.
- The authors needs to improve their Abstract. More specifically, please state the major unknown questions you are trying to solve, and provide sufficient information on dauricine and why you studied it.
Response: Thanks for the professional criticism, we rethought this question and re-edited the abstract. Thanks again.
- In the introduction, the authors claim “Microglia is the only resident immunoreactive cells in the brain”. This statement is not true, perivascular macrophages are also resident immune cells.
Response: Thanks for the professional criticism and correction, and we have replaced “the only resident immunoreactive cells in the brain” with “the main resident immunoreactive cells in the brain” to avoid dispute.
((1) Microglia have long been considered as the only immune cell type in parenchyma, while at the interface between CNS and the peripheral (meninges, choroid plexus, and perivascular space), embryonically originated border-associated macrophages (BAMs) and multiple surveilling leukocytes capable of migrating into and out of the brain. (PMID: 34881289); (2) Brain perivascular macrophages (PVMs) belong to a distinct population of brain-resident myeloid cells located within the perivascular space surrounding arterioles and venules. (PMID: 31749316); (3) Central nervous system (CNS) macrophages comprise microglia and border-associated macrophages (BAMs) residing in the meninges, the choroid plexus, and the perivascular spaces. (PMID: 32259484); (4) Brain-resident macrophages can be classified into microglia in the brain parenchyma and non-parenchymal brain macrophages, also known as central nervous system-associated or border-associated macrophages, in the brain-circulation interface. (PMID: 33972475)
- Figure 5 is hard to read. The authors may consider removing some panels (such as panel A) and make fronts on the others clearer.
Response: Thanks for your suggestion, and we have removed some panels and zoomed the others panels appropriately to make sure clearer to read.

Round 2
Reviewer 2 Report
The revised manuscript has addressed all my concerns and I am good with this being published.
Author Response
Thanks so much for your valuable time and professional suggestions, and thanks again for your contributions.